# A Quick Method to Synthesize Extrachromosomal Circular DNA In Vitro

**DOI:** 10.3390/molecules28104236

**Published:** 2023-05-22

**Authors:** Shanru Zuo, Xueguang Li, Yide Yang, Junhua Zhou, Quanyuan He

**Affiliations:** 1The Key Laboratory of Model Animals and Stem Cell Biology in Hunan Province, School of Medicine, Hunan Normal University, Changsha 410013, China; 2Department of Pharmacy, The Third Xiangya Hospital, Central South University, Changsha 410013, China

**Keywords:** extrachromosomal circular DNA, minicircle, PCR, ligase reaction

## Abstract

Extrachromosomal circular DNA (eccDNA) is a special class of circular DNA in eukaryotes. Recent studies have suggested that eccDNA is the product of genomic instability and has important biological functions to regulate many downstream biological processes. While NGS (Next-Generation Sequencing)-based eccDNA sequencing has led to the identification of many eccDNAs in both healthy and diseased tissues, the specific biological functions of individual eccDNAs have yet to be clearly elucidated. Synthesizing eccDNAs longer than 1 kb with specific sequences remains a major challenge in the field, which has hindered our ability to fully understand their functions. Current methods for synthesizing eccDNAs primarily rely on chemical oligo synthesis, ligation, or the use of a specific gene editing and recombination systems. Therefore, these methods are often limited by the length of eccDNAs and are complex, expensive, as well as time-consuming. In this study, we introduce a novel method named QuickLAMA (Ligase-Assisted Minicircle Accumulation) for rapidly synthesizing eccDNAs up to 2.6 kb using a simple PCR and ligation approach. To validate the efficacy of our method, we synthesized three eccDNAs of varying lengths from cancer tissue and PC3 cells and confirmed successful circularization through sequencing and restriction enzyme digestion. Additional analyses have demonstrated that this method is highly efficient, cost-effective, and time-efficient, with good reproducibility. Using the method, a well-trained molecular biologist can synthesize and purify multiple eccDNAs within a single day, and it can be easily standardized and processed in a high-throughput manner, indicating the potential of the method to produce a wide range of desired eccDNAs and promote the translation of eccDNA research into clinical applications.

## 1. Introduction

Circular DNA is widely present in nature, such as in bacterial genomic DNA, plasmids, mitochondrial DNA, chloroplast DNA, etc. Different from the above molecules, extrachromosomal circular DNA (eccDNA) is a special type of circular DNA which is derived from genomic segments of different chromosomes under certain conditions and which joins to form a ring in eukaryotes. It was first reported in 1965 [1]. Since then, researchers have found the existence of eccDNA in many species, such as yeast [2,3,4], nematodes [5], fruit flies [6,7,8], mice [9], humans [10,11,12,13,14], etc. Despite the widespread presence of eccDNA in eukaryotes, the research on eccDNA’s biological function has been stagnant for a long time due to the lack of suitable technology. Recent advances in rolling circle amplification sequencing (RCA-seq) have enabled researchers to globally identify and quantify the presence and abundance of eccDNAs at the genome level [10,12,15,16], providing a powerful tool for understanding the role of eccDNAs in cellular processes and diseases. It has been well-established that eccDNA is associated with human diseases [17,18] and many cancers [11,15,19,20,21,22,23,24,25,26,27,28]. The presence of eccDNA has been shown to promote the genetic heterogeneity of tumors [28], mediate intermolecular interaction and enhance oncogene’s expression [29], and correlate with genomic instability [22], which can all contribute to cancer development and progression [15,27,30]. By studying eccDNAs and their biological functions, researchers hope to gain a better understanding of the underlying mechanisms of these diseases and develop more effective strategies for diagnosis and treatment. However, the challenges in studying eccDNAs is their complex nature within cells. EccDNAs can exist in a range of sizes and structures, with variations in length, sequence, copy number, and chromosomal origin. Many recent studies have focused on identifying and characterizing populations of eccDNAs in different contexts, rather than investigating the functions of a single eccDNA molecule of a specific sequence. As a result, the mechanisms of how eccDNA interacts with other parts of the cell and its precise roles in cellular processes are still not fully understood.

One of the reasons for this knowledge gap is the lack of efficient and robust methods for synthesizing eccDNAs/minicircles with specific sequence in vitro. Most current methods for synthesizing eccDNAs rely on chemical oligo synthesis, DNA ligation [29,31], or require a specific gene editing and recombination system [32], which are limited in the length of the eccDNAs they can produce and are usually complex, expensive, and time-consuming. For example, Ligase-Assisted Minicircle Accumulation (LAMA) has been reported to synthesize 84–106 bp minicircle DNAs [33]. It uses two double-stranded DNAs that are complementary to each other in a split-reversed manner and requires several rounds of denaturation, annealing, and ligation to synthesize eccDNAs (Figure 1). Generally, these two linear DNA fragments were amplified by PCR based on two chemically synthesized oligos. As the synthesis of long DNA oligos (>100 bp) is still relatively expensive and time-consuming, this has limited its usage in the synthesis of longer eccDNAs. Later, a nontemplated ligation method was introduced which self-ligates the DNA oligomers of Hind III cohesive ends to minicircles via T4 DNA ligase [31]. However, this ligation reaction has two limitations. Firstly, the junction sites must be a HindIII cleavage site. Secondly, very low concentrations of oligonucleotides in the range of 0.1–10 nM are required in the ligation mixture to suppress inter-molecular ligation. To achieve a high yield, a large volume of ligation mixture and a substantial amount of T4 DNA ligase are required, which in turn limits the amount of production. To overcome the limitations mentioned above, Møller et al. [32] recently reported that CRISPR-C can generate eccDNA from intergenic and genic loci in a broad range of sizes from a few hundred base pairs and ranging up to a 47.4 mega base-sized ring chromosome. This system uses CRISPR-Cas nucleases to create double-strand breaks at desired sites in the genome, which can then be repaired through non-homologous end joining (NHEJ) or microhomology-mediated end joining (MMEJ) to generate circular DNA fragments. However, this method only works in vivo and requires a specific gene editing system, which is time-consuming and complex. Therefore, the development of a quick, efficient, and robust method to synthesize eccDNAs in broad length range is urgent in the eccDNAs research field.

Especially, the LAMA [33] approach is widely recognized as one of the most commonly utilized methods for synthesizing eccDNAs. It assembles a minicircle from two linear double-stranded DNA fragments which are complementary to each other in two half regions with reversed directions. The two fragments were mixed in a mole-based one-to-one ratio and subjected to multiple rounds of denaturing and annealing. As a result of this process, the fragments were able to partially anneal and form long overhangs at both ends, which were then converted into minicircles with two nicks. The minicircles were obtained by ligated-catalyzed ring closure via Taq DNA ligase. However, the original paper did not outline the procedure for synthesizing these complementary fragments. Typically, subsequent studies using LAMA methods relied on chemically synthesized oligos as templates to amplify target fragments via PCR. However, this approach is limited by the length of the oligos and can be both expensive and time-consuming (Figure 1).

In this study, we report a new method named QuickLAMA, which overcomes this limitation and has the capacity to generate spliced complementary DNA fragments for any genome region using regular PCR and ligation approaches. This method unlocks the full potential of the LAMA technique, enabling the high-yield synthesis of minicircles across a broader range of DNA lengths. Using this method, we synthesized three minicircles/eccDNAs of different lengths (from 731 to 2668 bp) in the microgram range within one day. The circularization of these eccDNAs were confirmed by gel electrophoresis, sequencing, and restriction enzyme digestion. The following analyses also suggest that the method is stable, simple, as well as efficient and will be beneficial for future eccDNA functional studies.

## 2. Results

### 2.1. Preparing Fragments A and E

We applied the QuickLAMA (see Methods section) to generate fragments A and E for three eccDNAs ranging from 731 to 2668 bp in length (lane 2 in Figure 2A–C) successfully, suggesting that the method is quite stable and easy to be applied.

### 2.2. The Synthesis of DNA Minicircle

Then, fragments A and E were mixed in a 1:1 molar ratio and went through the so-called LAMA reaction, which includes several rounds of the temperature program of denaturation (95 °C), annealing (4 °C), and nick ligation (65 °C) in a thermal cycler to accumulate circled DNA production. The products of the LAMA assay may contain the circularized DNA and remaining linear fragments A or E (lane 6 in Figure 2A–C). To remove these linear fragments, we utilized ATP-DNase to digest the LAMA products for 0.5–1 h and then purified the remaining minicircle using Cycle-Pure Kit (Omega). In order to evaluate the efficiency of DNA circularization in our method, we performed three technical replicates for all samples. The average concentrations of eccMir2392, eccBRCA1, and eccLIMD1 minicircles were 100.2 ng/µL, 90.5 ng/µL, and 148.2 ng/µL, respectively, in 50 µL of final eluted buffer. This means that a single round of QuickLAMA can yield approximately 5–7 µg of minicircles. Additionally, the low standard deviations observed in the concentration measurements for all three eccDNAs indicate that the method is highly reliable and robust (Table 1).

### 2.3. The Validation of DNA Minicircle Production

To validate the double-stranded circular nature of the products of the LAMA reaction, we utilized ATP-DNase to treat fragment A and the LAMA assay products and subsequently examined their digestion output via electrophoresis (lane 5, 7 in Figure 2A–C). As expected, the DNase treatment digested the linear DNA (fragment A) completely in all experiments (lane 5 in Figure 2A–C). For eccLIMD1, the products of the LAMA reaction contain one major band with similar molecular weight (MW) to fragments A and E (lane 6 in Figure 2C). The DNase-treated LAMA reaction products (lane 7 in Figure 2C) show a single band with higher molecular weight than fragment A and E. This result is consistent with the research results of Sunny et al. [34] that the migration rate of circular DNA was slower than that of the corresponding linear DNA molecules during agarose gel electrophoresis. The shift in molecular weight in the DNase-treated DNA may result from the binding of DNase to the DNA and was found in all experiments. As the DNA inputs of lane 6 and lane 7 were the same, the similar intensity of the two bands in these two lanes suggest the high efficiency of DNA circularization of eccLIMD1, which is close to 39.2% (Table 1). As for eccMir2392 and eccBRCA1, the products of the LAMA reaction contain two bands with similar intensity (lane 6 in Figure 2A,B). The band with a smaller MW was removed by the DNase treatment, suggesting that it contains mostly unligated linear DNA, and the efficiencies of DNA circularization are about 69.8% and 76.9%, respectively (lane 7 in Figure 2A,B and Table 1). After DNase treatment, both bands with the higher MW remained, suggesting the success of DNA circularization (lane 7 in Figure 2A,C). Additionally, we also cleaved the linear DNA (fragment A) and DNase-treated LAMA reaction products via two restriction endonucleases, which should produce three and two fragments with specific lengths (Figure 2A–C). As expected, the results of agarose gel electrophoresis (lane 8 and lane 9 in Figure 2A–C) show distinct patterns between linear and related circularized DNA and confirmed that the DNase-treated LAMA reaction products are minicircles. All of the bands shift slightly to higher MW in lane 9 of Figure 2A–C because of the DNase treatments. The junction sites of minicircles were amplified by inward PCR (Appendix A) and then sequenced using the Sanger method (Figure 2D–F). Very few point mutations were found in the body of the minicircles. Taken together, these results demonstrate that all three minicircles are generated correctly.

## 3. Materials and Methods

### 3.1. Experimental Design

The purpose of this study is to develop a quick method to synthesize eccDNAs of any specific sequences in vitro. The QuickLAMA method includes three steps: (1) the synthesis of two linear double-strand DNAs which can form the minicircles with two single-stranded nicks; (2) the use of LAMA to anneal and ligate nicks to form the minicircles (eccDNA) (Figure 1 and Figure 3); and (3) validation of the formation of eccDNAs via the digestion of plasmid-safe ATP-Dependent DNase enzyme and restriction endonucleases and the sequencing of the junction sites.

The key part of the method is synthesizing a dsDNA (fragment A and E) in step 1. The fragment E is spliced complementary to the related fragment A. The denatured single-stranded DNA (ssDNA) from the fragments A and E are capable of forming a duplex with approximate N/2 base pairs and two long cohesive ends of approximate N/2 nucleotides. The two long cohesive ends can then anneal to each other in the reverse direction to form minicircles containing two single-strand break sites (Figure 1).

### 3.2. Preparing Complementary Fragments (E) for the LAMA Reaction

Using phi29 DNA sequencing, we identified three target eccDNAs from a cervical cancer tissue and PC3 cells, which were named based on their proximity to neighboring genes. These eccDNAs are eccBRCA1 (731 bp), eccMir2392 (1135 bp), and eccLIMD1 (2668 bp). The loci of these eccDNAs are based on the hg38 reference genome and can be found in Table 2.

LAMA reaction required two linear double-strand DNAs which are complementary to each other in a split reversed way (Figure 1 and Figure 3). The target regions (fragment A) were amplified from human genomic DNA using regular KODFX PCR settings (Figure 3 step 1). The composition of 50 µL reaction buffer includes 1× KODFX PCR buffer, 2 mM dNTPs, 0.1 µM primer-1F, 0.1 µM primer-1R, 50 ng genomic DNA as the template, and 1 µL KODFX enzyme. The mix was heated to 95 °C for 2 min. Then DNA was denatured at 98 °C for 10 s, annealed at 52 °C for 30 s, extended at 68 °C for 1–4 min for 30 cycles, and chilled at 4 °C.

Then, four PCR reactions were performed to generate four DNA fragments (C-long, C-short, D-short, and D-long in Figure 3 step 2), which covered two parts (Part C and Part D) of the target region (fragment A) in two long and short forms. Specifically, the primer 2R has three additional bases which complement the 5′ end of the primer 1F to form a stick end (see below). These PCR assays took fragment A as the template and used a similar KODFX mixture and PCR program. The purity and concentration of PCR products were checked by electrophoresis and Nanodrop one.

The long and short double-stranded DNA (dsDNA) fragments of each part (C/D) (approximately 500 ng) were mixed in a 1:1 molar ratio and then denatured at 95 °C for 2 min and annealed by decreasing the incubation temperature 5 °C per 3 min to 30 °C. During the annealing step, theoretically, about 50% of ssDNA from long and short fragments may form hybrid dsDNA fragments with one stick and one blunt end. After being treated with T4 polynucleotide kinase (Figure 3 step 3), the hybrid dsDNAs from C and D parts were mixed with 1:1 molar ratio and ligated by T7 kinase enzyme which specifically catalyzes the ligation of cohesive ends but not blunt ends. The composition of the 20 µL reaction buffer includes 2× T7 DNA ligase reaction buffer 10 µL, C-sticky DNA 4.5 µL, D-sticky DNA 4.5 µL, and 1 µL T7 DNA ligase. This step enables the specific ligation of the stick ends from the two ends of fragments C and D and suppressed the inter-molecular ligation (Figure 3 step 4).

Then, using the ligation products as templates, the fragment E can be amplified by PCR with specific primers. The composition of 50 µL reaction buffer includes 1× KODFX PCR buffer, 2 mM dNTPs, 0.1 µM primer-3F, 0.1 µM primer-3R, T7 ligated sticky C and sticky D as a template, and 1µL KODFX enzyme (Figure 3 step 5). All PCR primers were ordered from Tsingke Biotechnology Co., Ltd. (Changsha, China) (Appendix A). The PCR procedure is the same as the previous one. The sequences of the PCR products (fragment E) can be validated by Sanger sequencing if needed.

### 3.3. DNA Circularization by LAMA Method

The procedure of the LAMA method was previously reported [33]. Fragments A and E were purified with E.Z.N.A. Cycle-Pure Kit (Omega, Norcross, GA, USA) according to the manual to desalt and remove enzymes and contamination, and then phosphate groups were added to the 5′ end of these fragments by T4 PNK kinase (NEB) at 37 °C for 30 min following inactivation at 65 °C for 20 min. About 500 ng of fragment A and E were mixed in 5 µL 10× HIFI Taq DNA ligase buffer with 1 µL of HIFI Taq DNA ligase (NEB) for the Ligase-Assisted Minicircle Accumulation (LAMA) procedure. The mixture was placed in a thermal cycler (BIO-RAD T100 thermal cycler) that completes the following temperature program: step 1, DNA denaturation at 95 °C for 20 s; step 2, cooling at a maximum rate of 4 °C for 1 min; step 3, ligation at 65 °C for 20 min for 3–10 cycles. The DNA circles ranging from 700 bp to 3 kb were produced by this procedure.

### 3.4. The Validation of Circularization Products

We used three methods to validate the formation of the minicircles/eccDNAs: (1) The products of LAMA reaction were treated by plasmid-Safe ATP-dependent DNase (Epicentre, Madison, WI, USA) to check its resistance to the DNase digestion. The composition of 50 µL reaction buffer includes 600 ng target DNA, 2 µL 25 mM ATP, 5 µL 10× Reaction Buffer, and 1 µL Plasmid-Safe DNase. After incubation at 37 °C for 2 h, the DNase was heat inactivated for 30 min at 70 °C. (2) The double-stranded circular nature of the products was verified via digestion with two restriction endonucleases by comparing the patterns of digested fragments of eccDNAs and their linear counterparts using electrophoresis. (3) The minicircles were sequenced by the Sanger method to validate their connection. The sequencing results were analyzed by Snap-Gene Viewer.

To further evaluate the efficiency of the circularization, the products of LAMA reaction were digested by plasmid-Safe ATP-dependent DNase (Epicentre) to remove linear DNA. The purity of these products was checked by 1.5–2% agarose gel electrophoresis and quantified by NanoDrop. The efficiency of the circularization was measured by the ratio of the amount of DNase treated DNA to the input DNA. The eccDNA products of QuickLAMA often exhibit multiple bands or a smear of big bands when run on regular agarose gel conditions, making it challenging to evaluate the synthesis results. To address this issue, we have added 0.5% SDS (sodium dodecyl sulfate) to the DNA loading buffer to eliminate DNA–protein interactions and prevent the appearance of additional bands due to the diverse topology of DNA circles.

## 4. Discussion

Advances in sequencing technologies have allowed the identification and mapping of eccDNAs at the genome scale. It is now clear that the populations of eccDNAs are extremely diverse across tissue and cell lines. Although accumulated evidence highlights that eccDNA plays important roles in many cellular processes and human diseases, it is still controversial whether unique types of eccDNA play significant roles in cellular processes and human disease progression. The understanding of single eccDNA is still very limited and has been hindered by the lack of an efficient method to synthesize DNA minicircles in vitro.

QuickLAMA provides a simple and robust way to solve this problem. Compared with other methods, it has several advantages: (1) QuickLAMA is time- and cost-efficient. The whole procedure of QuickLAMA can be performed within one day in most molecular biology labs with a PCR machine; for each minicircle, only six customized primers and several regular enzymes are needed. (2) The yield of QuickLAMA is high. As fragments A and E can be easily amplified by PCR and the efficiency of LAMA is quite high, it is easy to synthesize minicircles in the microgram to milligram level. (3) QuickLAMA can synthesize minicircles in a wider range of length. The only limitation of the length of minicircles in QuickLAMA may be derived from the PCR method, which can usually amplify DNA fragments of between 0.1 and 10 kb in length. (4) As QuickLAMA does not depend on any specific sequences, it is easy to add customized sequences or chemical modifications in minicircles. (5) As all steps in QuickLAMA can be standardized, it is easier to extend it to a high-throughput level. All of these features make QuickLAMA a powerful method for the scientific study of eccDNAs and the clinic usage of minicircle.

Additionally, QuickLAMA also has some limitations. For example, it may be difficult for QuickLAMA to synthesize minicircles longer than 10 kb (such as eccDNAs in cancer cells, which are usually longer than 1M bp). Although some PCR techniques allow for the amplification of fragments up to 40 kb, whether they are compatible with our method needs to be further validated. Secondly, in some cases, the junction sites with low sequence complexity may make it difficult for PCR to amplify target fragments, which, fortunately, can usually be solved by an adapter or PCR condition optimization. Additionally, the current QuickLAMA protocol was only validated for eccDNAs originating from single continuous genomic loci (sl-eccDNA) in this work but not for ones from multiple genomic loci (ml-eccDNA). However, using QuickLAMA to synthesize ml-eccDNA should be straightforward to achieve by including an extra step to ligate the targeting genomic fragments and form the fragment A. Finally, it should be noted that DNA minicircles synthesized by QuickLAMA may not be biologically identical to their natural counterparts (eccDNA) due to the absence of histone octamers and related histone or DNA modifications.

## 5. Conclusions

In conclusion, QuickLAMA is a simple and reliable approach for synthesizing minicircles ranging from a few hundred to several thousand base pairs in vitro. This method promises to ease the functional analysis of eccDNA and unlock the full potential of minicircles for clinical applications.

## Figures and Tables

**Figure 1 molecules-28-04236-f001:**
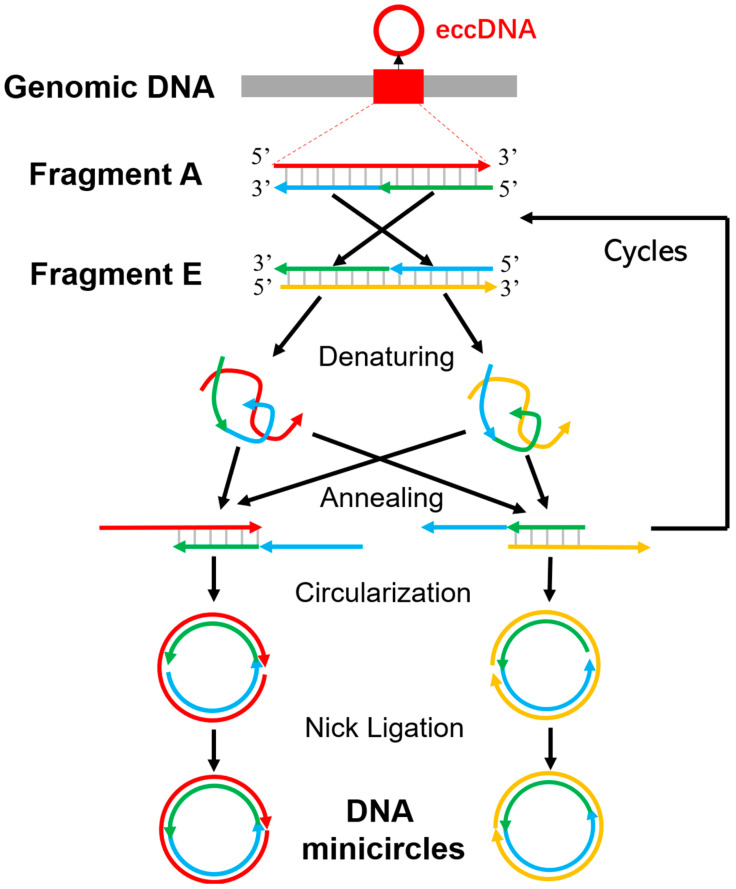
The procedure of the LAMA protocol. The top panel shows two DNA duplexes (Fragment A and E) which are complementary to each other in two half regions with reversed directions. They are mixed with a 1:1 molar ratio and undergo several denaturing and annealing cycles. During these cycles, a portion of the oligos from the two DNA duplexes can anneal to create fragments with two long overhangs, which promotes the circularization. The remaining oligos return to linear duplexes and may produce circularized DNA duplexes in subsequent rounds. Finally, Taq DNA ligase is used to ligate the nicks of the circularized DNA duplexes.

**Figure 2 molecules-28-04236-f002:**
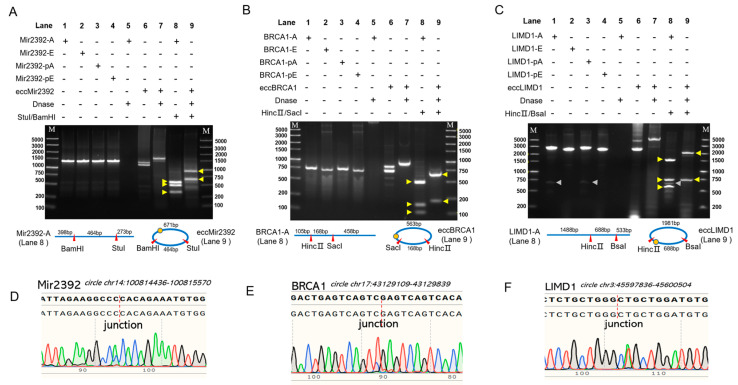
The validation of the three synthesized minicircles. (**A**–**C**). The agarose gel electrophoresis results of the intermediate/final products of QuickLAMA experiments for eccMir2392, eccBRCA1, and eccLIMD1. Lanes 1 and 2 are fragments A and E of the targeted eccDNA; lanes 3 and 4 are the phosphorylated fragments A and E; lane 5 shows the products of DNase-treated fragment A; lanes 6 and 7 are the products of the QuickLAMA assay without and with DNase treatment; lanes 8 and 9 are the products of double-restriction enzyme digestion of fragment A and DNase-treated QuickLAMA products. The patterns of double-restriction enzyme digestion in linear DNA (lane 8) and related minicircles (lane 9) are illustrated in the bottom panels. The yellow arrows indicate the fragments resulting from restriction enzyme digestion. The grey arrows in (**C**) indicate the unspecific bands inherited from fragment (**A**).(**D**–**F**). The sequencing results of the junction sites of the three minicircles. The red dashed lines indicate the junction sites.

**Figure 3 molecules-28-04236-f003:**
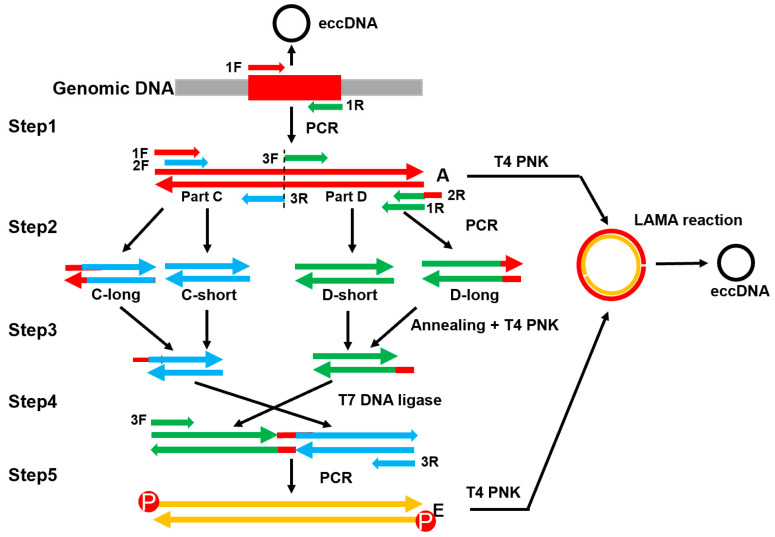
The schematic illustration of the QuickLAMA. There are five steps in the QuickLAMA protocols. The red region in the genomic DNA is the locus that was circulated to form eccDNA. All primers used in the protocols were represented as small arrows and labeled with specific tags. The detailed conditions of all steps were described in the main text and Supplemental Materials (Appendix A).

**Table 1 molecules-28-04236-t001:** The evaluation of the products of the QuickLAMA method (mean ± standard deviation).

eccDNA Name	Circularization Ratio (%)	Con. (ng/µL)	A260/280	A260/230	Yield (µg)
eccBRCA1	76.93 ± 9.67	101.1 ± 1.80	1.82 ± 0.01	2.43 ± 0.25	~5
eccLIMD1	39.22 ± 17.69	143.6 ± 2.45	1.87 ± 0.01	2.34 ± 0.25	~7.15
eccMir2392	69.79 ± 5.26	100.77 ± 1.93	1.87 ± 0.03	2.13 ± 0.14	~5

**Table 2 molecules-28-04236-t002:** Three targeting eccDNAs used in the study.

eccDNA Name	Chromosome	Start	End	Length	Description
eccBRCA1	chr17	43,129,109	43,129,839	731	Upstream of BRCA1 gene
eccLIMD1	chr3	45,597,836	45,600,504	2668	The first intron of the LIMD1 gene
eccMir2392	chr14	100,814,436	100,815,570	1135	Containing mir2392 gene

## Data Availability

The data presented in this study are available in Appendix A.

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
