# Peer review of "A Quick Method to Synthesize Extrachromosomal Circular DNA In Vitro"

_molecules, 2023, doi:10.3390/molecules28104236_

Round 1
Reviewer 1 Report (Previous Reviewer 3)
I have reviewed the authors' second response to my original review. I pointed out previously that the absence of sets of topoisomer bands in Figure 3 A-C raised serious concerns about the assignment of band that is attributed to the main product. In response, the authors have submitted a set of re-annotated gel images and an explanation. It is claimed that an additional minor band noted in two of the gels addresses previous concerns about the absence of an equilibrium topoisomer population.
The data are unconvincing. In the case of the largest species, only two bands are visible: one labeled supercoiled and the other labeled "relaxed coil." More than two bands are expected for circles in this size range. It is also unclear whether this "relaxed coil" species is meant to refer to covalently closed DNA with a Delta (Lk) near 0 or the open-circular form. If it is the latter, then the relaxed coil state of the eccBRCA1 circle has been incorrectly assigned.
There is a rich and classic literature on the gel-electrophoretic properties of circular DNA in general and DNA minicircles in particular. A more rigorous analysis of the properties of these eccDNA mimics is warranted.
Author Response
Thank you for your comments. We believed that the "relaxed coil" species is covalently closed DNA, otherwise it can’t be resistant to the DNase digestion (lane 7).
We are uncertain as to why the number of bands observed is smaller than what the reviewer expected. Is it possible that the DNase-DNA binding has resulted in a simpler pattern? In fact, we consistently observed two bands in each sample, but their relative abundance varied in different repeats. This suggests that the use of the topoisomer bands pattern alone may not be a reliable indicator of circularized DNA.
To our knowledge, the most reliable way to identify circular DNAs is full length and junction site sequencing. And our sequencing results supported the minicircle DNA nature of QuickLAMA products. Furthermore, the restriction enzyme digestion experiments show clear different patterns between eccDNAs and their linear counterparts in all three samples, which further validate the nature of circular DNA of QuickLAMA products; Additionally, the reality that the products of QuickLAMA are resistant to the digestion of plasma safe DNAase also supports they are not linear DNAs. If there are more rigorous analysis methods available, please let us know, Thank you.
Reviewer 2 Report (Previous Reviewer 2)
Zuo S. et al. attempted to answer all the issues raised in the first revision and operated all the suggested corrections to the manuscript "A Quick Method to Synthesize Long Extrachromosomal Circular DNA in vitro". The quality of the manuscript has improved both in terms of form and content.
At this point, it is up to the Academic Editor whether to accept the publication of this paper.
Thank you very much.
Author Response
Thank you for your valuable suggestion and comments.
Reviewer 3 Report (Previous Reviewer 1)
Everything has been addressed
Author Response
Thank you for your valuable suggestion and comments.
Reviewer 4 Report (New Reviewer)
In their manuscript, Zuo and colleagues describe a ligase-assisted LAMA-based method for the production of minicircles ex vivo, aiming to create an efficient easy-to-perform platform for the study of individual eccDNAs. Methodically, mixture, denaturation and ligation of two linear double-stranded target DNAs that are complementary to each other in a split-reversed fashion leads to circle formation in presence of a cohesive end-sensitive ligase. Products can be detected by PCR and validated by Sanger sequencing spanning the junctional regions.
In principle, the authors address a relevant gap in current circular DNA researches. Though the knowledge on eccDNA structure, amplicon rewiring and role in circular and linear gene expression and transcription changes has tremendously grown, modes for functional characterization of individual target eccDNAs are underrepresented. In their protocol, the authors address this issue, which might promote the aim to define a prognostic or diagnostic biomarker role of eccDNA in diverse disorders, also beyond oncology. However, I see several critical issues the authors should consider:
Major comments:
- General: the article would profit from an English editing service to improve language usage.
- Citations need fundamental re-organization and control:
- Introduction, lines 35-37: …‘in most of cell lines, tissues and organs across species in eukaryote’. The literature cited here (ref. 1-6) comprises heterogeneous studies performed in human cancer types, sperm, muscle, and in yeast and plants. Conditions are not directly comparable and therefore need to be outlined more in detail.
- Introduction, line 36: ‘Free from the chromosomal genome’…meaning is unclear as eccDNA is of genomic, non-mitochondrial, non-plasmid origin.
- Introduction, line 38: Ref. 7 is not appropriate, maybe can be replaced by ref. 5.
- Introduction: What is the purpose of the authors to cite ref.11-15? These studies do not exclusively ‘identify and quantify eccDNA at the transcriptome level’ but implicate a characterization at the genome level. eccDNAs can arise from genic, inter-genic and non-genic regions and thus from coding and non-coding regions.
- Introduction, line 40: ‘RC-seq’, this term should only be used if coined by the ref. cited (ref. 8-11).
- Introduction, line 44: This statement is not supported by current research as a concrete role of eccDNA in neurodegenerative disorders has not yet been demonstrated. Ref. 16 (Helmsauer et al.) is on neuroblastoma, a tumor arising from sympathetic neurons of the adrenal gland, which does not belong to the scope of neurodegenerative diseases. Wang is incorrectly cited as the study is not carried out in the context of an autoimmune disorder. A role of ecDNA has mainly been demonstrated in tumors rather than in other disorders. Instead of ref. 17 and 22, the authors should cite original work from pioneering groups.
- Introduction, lines 45/46: Sentence is imprecise. Better, e.g.: ‘genetic heterogeneity of tumors.’ Ref. 6 is carried out in plants thus appears inappropriate in this context. Ref. 12 (Shibata et al.) appears incorrectly cited as eccDNA is a product of genomic instability, not a cause of it.
- Introduction, line 47: Ref. 13,14,16 give deeper insight into the regulation of ecDNA in tumors but no other diseases. The tumor context and the role of circular DNA in intermolecular interaction and amplification of oncogene expression and transcription should be specified.
- Introduction, line 52/53: ‘Many recent studies have focused on identifying and characterizing populations of eccDNAs in different contexts, rather than investigating the functions of individual molecules.’ This sentence neglects the quantity of high impact publications that characterized the role of circular DNA in tumor genetics and clinical outcome and prognosis (see, e.g., Koche et al., Nature Genetics, 2020; Kim et al., Nat Gen, 2020).
- Methods, lines 118-123 etc: It is not getting clear whether QuickLAMA has advantages or disadvantages compared to the original LAMA protocol (ref. 25) beyond the capacity to broaden the size of minicircles synthesized. Though the application of their method is well-exemplified for three eccDNA sequences, a direct comparison of key output parameters (efficiency, template specificity, purity, size range) would help scientists in their choice to use this protocol. Moreover, the protocol would substantially gain from detailing or even exemplifying the option to generate circles with multiple genomic origins.
- Results, line 205-213: This passage summarizes the principles of the technique and therefore should appear as a central element of the Methods part, but be removed from the Results section
- Is there any scientific indication for the assumption that DNase binding might be responsible for the considerable shift in MW of the circular products in Fig. 3A-C?
- Fig. 3: Why, in A-C, is the band for the purified eccDNA product (lane 7) stronger than for the non-purified product (lane 6)? The visual results are thus discrepant to the description in the text.
- Results, line 252: After DNase treatment, the stronger band in lane 6 of Fig. 3C is disappearing. Thus, how can the output of purified circularized DNA (for LIMD1 in lane 7) be higher than 50%?
- The authors miss to discuss how their minicircles can be introduced into cellular systems. The protocol would greatly profit from demonstrating that artificial minicircle accumulation conveys a biological function. How stable are the circles intracellularly? What is the rough number of identical circles that make up an average conc. of 100ng/μl per QuickLAMA preparation?
Minor comments:
- Abstract, line 13/14: replace ‚procedures‘ by ‚processes’
- Abstract, line 19 onwards: ‘or the use of specific (without ‘a’) gene editing or recombination systems’. I suggest to replace by: ‘These methods are often complex, time- and cost-intensive and limited by the length of eccDNAs generated
- Abstract, line 23: add space between ‘up’ and ‘to’
- Abstract, line 30: ‘mannel’? I recommend to rephrase and suggest, e.g.: ‘indicating the potential of the method to produce a wide range of desired eccDNAs and promote the translation of eccDNA research into clinical applications’
- Keywords: delete: ‘Synthesize’
- Introduction, line 35-37: This sentence is grammatically incorrect and should be rephrased
- Introduction, line 43: Start new sentence with upper-case letter
- Introduction, line 62: ‘Ligase-Assisted Minicricle Accumulation (LAMA)‘, correct to ‚Minicircle’
- Introduction, line 68/69: correct to ‘was introduced’
- Introduction, line 82: correct to ‘which is time-consuming’…CRISPR-Cas9 system is functional in cell lines and thus further CRISPR-C developments might principally be an option also in in vitro systems
- Introduction, line 76, 248: ’Møller, Henrik Devitt et al…’ and ‘results of Sunny Shin et al’…cite last name only
- Introduction, line 113: what kind of analyses?
- Nomenclature: The authors should distinguish between longer Mb ecDNAs and double minutes found in tumors, and smaller eccDNAs
- Methods, line 122-123: ‘Validation of the formation’…
- Methods, lines 123, 298: ‘junction sites’ instead of ‘join sites’
- Methods, line 124, heading: why ‘(E)’, if the technique can be used for chromosomal DNA fragments of any position in the genome?
- Methods, line 132: reduce to ‘Figure 1 and 2’
- Methods: ‘μM’, ‘μl’ etc.: all or none of the measurements in italics
- Figure 2: details given in the text such as ‘Figure 2 step1’, Figure 2 Step2’, Part/part, fragment etc. should consistently appear either with upper-case or lower-case letters
- Methods, lines 147, 152: ‘1:1 molar ratio’
- Methods, line 155: replace ‘u’ by ‘μ‘
- Methods, line 163: ‘the sequences of the PCR products’
- Methods, line 175: ‘following inactivation’?
- Methods, line 190: ‘by the digestion WITH 2 restriction endonucleases’?
- Results, line 204: add space between section number and ‘Results’
- Results, line 219: ‘fragment E’ instead of ‘fragment Es’
- Results, line 222: After the heading 3.2. there is a structural gap, and also the text suggests that a passage is missing. Please check
- Results, line 231/232:…’The average concentrations were…’; ‘respectively’ between commata
- Results, line 233: please correct: …’can yield’… and: ‘of minicircles’
- Results, line 246/247: The shift of MW for the eccDNA products are present in A-C, not only in C, as stated thereafter. Line 250: ‘molecular weight’ instead of ‘molecule weight’
- Results, line 257: ‘unlighted’? or ‘unligated’
- Results, line 258: …’respectively’….
- Legend of Figure S1: Correct to: ‘Schematic illustration of the inward PCR primer design’
- Table S2 is misplaced and should be centered when displayed on a page
- Results, line 269, line 274: separate 3D-F with spaces; space after comma (in the brackets)
- Discussion, line 283 onwards: lower case letter after semicolon
- Discussion, line 285/296: consistently either kb or kbp
- Figure 3. ‘A-C’ instead of ‘A~C’; line 325: correct to: ‘Lane 5 shows the products of DNase treated A fragment.’ ‘DNase’ instead of ‘Dnase’; line 331: ‘Figure E’ instead of ‘figure e’. Grey arrows are given in Figure 3C, not 3E. What is the entity of the additional products in Figure 3A, lines 1-4?
- Abbreviations: list is incomplete, e.g., add ssDNA, dsDNA, MW, NHEJ, MMEJ etc.
- Figure 2: ‘genomic DNA’ instead of ‘genome DNA’
- Table S3: Step1, left: correct to ‘genomic DNA’; step 2, left: ‘DNA’ instead of ‘DAN’; step 3, ‘inactive condtion’, replace by ‘inactivation’; time: ‘1-3h’ instead ‘1~3h’; description, step1: replace ‘in the Figure 2’ by ‘(see Figure 2)’; description, step 2: ‘obtain’ instead of ‘got’; description, step 3: correct to: ‘Ligation of the fragments to form the fragment E as the template for the next step’; Step 5, 7: ‘reaction buffer’ (add space); ‘DNase’ instead of ‘Dnase’; Step 4: ‘needs to be diluted’; Step 8: delete surplus space between ‘purify’ and ‘the’.
Author Response
Please see the attachment

This manuscript is a resubmission of an earlier submission. The following is a list of the peer review reports and author responses from that submission.
Round 1
Reviewer 1 Report
The authors present an adaptation of a previously published method to adapt ligase mediated DNA mini circle formation to the use of PCR templates instead of presynthesized oligo blocks. The authors propose using this technique to make eccDNA of known sequence to study the function of specific eccDNA. In principle this method could be important to the field, however the current manuscript is held back by several limitations.
Results: The authors state that the efficiency was measured, but no presentation of the efficiency was given. No indication on the reproducibility was given either. That is, is the efficiency constant and consistent at least for the molecules tested?
The method described is woefully undetailed and unclear. Part of this is due to the extensive english language editing required, but part also due to a lack of clarity. This manifests itself in several instances:
1.) Figures 1 and 2 should both be referenced in the method around line 74. Figure 2 is much more clear than figure 1 in generating circles and should precede the PCR schema.
2.) Sequencing of the joints was not described. What were the sequencing primers used? Were there any mutations in the rest of the circles observed? Did the authors sequence the entire circles produced to verify they were in fact the intended product and not just contain the junction?
3.) Table 1. What genome build are the coordinates for the eccDNA given from? HG19? HG38? Did the authors conduct the phi29 sequencing for these or were they pulled from publicly available data? If so what are the accession and availability of these identified circles?
4.) Line 89 “were mixed 1 to 1 ratio” is this based on the MW of the fragment or the ul of the reaction added? If this protocol were to be performed by others on other eccDNA its conceivable the long and short portions of C and D fragments could be different enough lengths that require equimolar instead of equivolume. Please clarify. This is true for each instance of mixing 1 to 1.
5.) Line 108-109. Fragments A and E were purified: is this from the gel by cutting out only the correct sized product, or straight from the reaction.
6.) The authors describe validation of the circularization products (line 119 on) however part of this would be in completing synthesis of the products. The authors should more clearly describe what they would do to complete the generation of pure single eccDNA ready for the addition to cells.
The introduction could lay out the question and the problems with existing techniques better and why QuickLAMA is an improvement in a more clear way. For instance, most casual readers will not realize why you can’t just ligate the ends of a standard PCR product to make the circle. Why is intermolecular ligation a problem in this? Why QuickLAMA avoids these issues and promotes circular products and not concatenation is sorely necessary.
Author Response
1.) Figures 1 and 2 should both be referenced in the method around line 74. Figure 2 is much clearer than figure 1 in generating circles and should precede the PCR schema.
Re: Thank you for the suggestion. We switched the order of Figure 1 and Figure 2 in the revised manuscript. And the figure 1 has been redrawn to make the circularization procedure clearer.
2.) Sequencing of the joints was not described. What were the sequencing primers used? Were there any mutations in the rest of the circles observed? Did the authors sequence the entire circles produced to verify they were in fact the intended product and not just contain the junction?
Re: We sequenced the junction sties of DNA minicircle by designing a pair of inward sequencing primers. We added a schematic diagram of designing inward PCR primers in Figure S1and the sequences of the primers in the Table S1. For shot eccDNAs (<1000bp), The two inward sequencing results can cover all sequence of eccDNAs. For longer ones, for example eccLIMD1, we did full-length sequencing with additional primers and the results show expect sequences. We did observe very few point mutations in eccDNA product when mapping them to reference genome, which may from SNPs or PCR errors and should have very minor effects on the eccDNAs. Additionally, beside the sequencing, we also used the restrict enzymes to cut the minicircles to verify the generation of the intended products.
3.) Table 1. What genome build are the coordinates for the eccDNA given from? HG19? 38? Did the authors conduct the phi29 sequencing for these or were they pulled from publicly available data? If so what are the accession and availability of these identified circles?
Re: The gene information in Table 1 is based on hg38. The three eccDNAs were identified from a cancer tissue sample and PC3 cell line using round rolling circle amplification sequencing (RCA-Seq) from one of our ongoing projects and the data has not yet been published. We added these details in the revised manuscript. Please refer to the Table 1.
4.) Line 89 “were mixed 1 to 1 ratio” is this based on the MW of the fragment or the ul of the reaction added? If this protocol were to be performed by others on other eccDNA its conceivable the long and short portions of C and D fragments could be different enough lengths that require equimolar instead of equivolume. Please clarify. This is true for each instance of mixing 1 to 1.
Re: Thank you for the question. Here we are using a molar ratio of 1:1 for mixing. We have provided clarification and additional information in the article (line 225).
5.) Line 108-109. Fragments A and E were purified: is this from the gel by cutting out only the correct sized product, or straight from the reaction.
Re: Fragments A and E were purified using straight from the reaction. We only used gel to do QC of these PCR products.
6.) The authors describe validation of the circularization products (line 119 on) however part of this would be in completing synthesis of the products. The authors should more clearly describe what they would do to complete the generation of pure single eccDNA ready for the addition to cells.
Re: The products of LAMA assay may contain the circularized DNA and remained linear fragment A or E (lane 6 in Figure 3A, 3B, and 3C). To remove these linear fragments, we utilized ATP-DNase to digest the LAMA products for 0.5~2h and then purified remained minicircle using Cycle-Pure Kit (Omega) which is ready for the addition to cells. We added this details in the section of “The synthesis of DNA minicircle”. We also added a comprehensive protocol of QuickLAMA in supplemental data. Please refer to the line 229 and supplemental data.
If you have additional suggestions and comments about the manuscript, please let us know, we will try our best to improve it.
Thank you so much!
Shanru Zuo
Hunan Normal University
Changsha, Hunan, CHINA
2/26/2023

Reviewer 2 Report
The article “A Quick Method to Synthesize Long Extrachromosomal Circular DNA in Vitro” by Zuo S. et al. is an experimental work that investigates a possible efficient and stable method to synthesize individual eccDNA in vitro, i.e., a novel method named by the authors as QuickLAMA to synthesize long minicircles using basic PCR and ligation assay for potential clinic applications.
However, the article has many shortcomings as follows:
1. The title is not correctly written in the all-caps title style (please, see correction in the title, stipulated in RED) and does not accurately illustrate the experiment performed. The title should have been more carefully chosen.
2. For the keywords: The authors did not adhere to MeSH for choosing keywords and have several imperfections in keyword formulations.
3. The editing was done in hurry and with mistakes, for example:
In a lot of lines, the pause before the cited reference is missing! For example, incorrect in line 25: “pathological processes in special ways[1-3].” Please, correct as:
“pathological processes in special ways [1-3]”- a space before the square bracket is required!
And the examples can go on: lines: 26, 27 ,34, 35, 36 etc... – in the whole manuscript!!!
4. A better representation of the experiments and the work is necessary for the reader, with a control method to prove the results!
5. Article has some inconstancies, I think additional experiments are needed to control the results obtained, and to prove that the experimental work was conducted rigorously.
6. More comprehensive arguments and explanations on the presented results in the Figures would be very welcome.
7. The manuscript is missing important information, and it could be improved in Results and Discussions.
8. More comparative arguments regarding the actual results obtained are missing!
9. Conclusions are NOT objective, they seem rather formulated as the authors wanted, that is, they are subjective!
10. References are too few and are not written in the required MDPI / Molecules format, for example the number of volume should be in Italics. All references must be rewritten correctly and double-checked.
11. A “List of Abbreviations” must be completed and reviewed carefully and may be better presented in a table format in the end.
12. Extensive editing of English language and style are required.
13. Overall, I recommend a carefully revision!
14. I believe that after this revision provided by the authors on the issues suggested to be corrected and improved, it will provide useful and credible information for all readers and especially researchers and it is up to the Academic Editor to decide on its publication.
Thank you very much!
Author Response
1. The titleis not correctly written in the all-caps title style (please, see correction in the title, stipulated in RED) and does not accurately illustrate the experiment performed. The title should have been more carefully chosen.
Re: Thank you for pointing out the mistake. It has been corrected in the revised paper.
2. For the keywords:The authors did not adhere to MeSH for choosing keywords and have several imperfections in keyword formulations.
Re: Thank you for the suggestion. We changed some of keywords in the revised paper and made sure that all of them can be found in the MeSH database.
3. The editing was done in hurry and with mistakes, for example:
In a lot of lines, the pause before the cited reference is missing! For example, incorrect in line 25: “pathological processes in special ways[1-3].” Please, correct as:
“pathological processes in special ways [1-3]”- a space before the square bracket is required!
And the examples can go on: lines: 26, 27 ,34, 35, 36 etc... – in the whole manuscript!!!
Re: Thank you very much. We have fixed these issues and checked the whole manuscript carefully to avoid the writing mistaken.
4. A better representation of the experiments and the work is necessary for the reader, with a control method to prove the results!
Re: We added more background and details to improve the representation of the experiment. For example, figure 1 has been redrawn to make all steps of LAMA clearer. We also added two new tables about the origin of the targeting eccDNAs and the evaluation results of QuickLAMA products. Additionally, a detailed table formatted protocol of QuickLAMA has been added to the supplemental data. We used DNA circular junction sequencing to confirm the formation of junction site, and digestion with ATP-Dnase enzyme as well as restriction endonucleases targeting each DNA circle to digest both the DNA circles and the corresponding linear DNA (control), followed by gel electrophoresis to verify the formation of DNA circles.
5. Article has some inconstancies, I think additional experiments are needed to control the results obtained, and to prove that the experimental work was conducted rigorously.
Re: Not sure what the inconstancies are. But we did additional experiments such as full-length sequencing to make sure the long eccDNAs have the correct sequences. We also provided more comprehensive results including all primer sequences (Tabel1 in supplemental data) and the concentrations, purity and yield of the products/eccDNAs in our study to prove the rigorousness of our experiments, please refer to the table 2.
6. More comprehensive arguments and explanations on the presented results in the Figures would be very welcome.
Re: Thank you for pointing it out. We added more comprehensive details in the text and figure legend to make it easier for reader to interpret out results. Please refer to Figure 1, Table 1 and Table 2.
7. The manuscript is missing important information,and it could be improved in Results and Discussions.
Re: We revised the introduction and results.
8. More comparative arguments regarding the actual results obtained are missing!
Re: As mentioned before, we improved the introduction, results sections by adding more details about the experiments or the protocol in the revised manuscript. New data about circularization ratio, concentration, purify and yield of the product were provided in the revised manuscript. Please refer to Table 1, Table 2 and Table S1.
9. Conclusions are NOT objective, they seem rather formulated as the authors wanted, that is, they are subjective!
Re: Thank you. We rewrite the conclusions section and moved some subjective statements to the discussion section.
10. References are too few and are not written in the required MDPI / Molecules format, for example the number of volume should be in Italics.All references must be rewritten correctly and double-checked.
Re: Thank you. We added more relevant references. The references format were checked and corrected carefully. Please refer to the reference section in the revised paper.
11. A “List of Abbreviations” must be completed and reviewed carefully and may be better presented in a table format in the end.
Re: We added “List of Abbreviations” at the end of the revised manuscript.
12. Extensive editing of English language and style are required.
Re: We asked a native speaker to polish the manuscript.
13. Overall, I recommend a carefully revision!
Re: Thanks for your suggestion, we tried our best to improve the manuscript.
14. I believe that after this revision provided by the authors on the issues suggested to be corrected and improved, it will provide useful and credible information for all readers and especially researchers and it is up to the Academic Editor to decide on its publication.
Re: Thank you for your patient review and constructive suggestion.
If you have additional suggestions and comments about the manuscript, please let us know, we will try our best to improve it.
Thank you so much!
Shanru Zuo
Hunan Normal University
Changsha, Hunan, CHINA
2/26/2023
Reviewer 3 Report
This manuscript describes an in-vitro synthesis method for analogs of extrachromosomal-circular DNAs based on an extension of Ligase-Assisted Minicircle Accumulation (LAMA). LAMA was first described in a 2008 paper by Du et al. (Nucleic Acids Res 36, 1120-1128). The technique described here appears to modestly advance the original implementation of the LAMA method by eliminating a need for long synthetic oligonucleotides. Instead, only three pairs of eccDNA-specific primers and the original eccDNA template are needed to carry out the synthesis.
Given how incremental the advance, the manuscript seems better suited to a methods-oriented journal. In addition, I have several reservations:
1. The title of the manuscript is misleading. eccDNA sizes can range into the hundreds of kbp or even Mbp. The authors have demonstrated the utility of their method only up to 2.7 kbp, which is possibly near the average of many eccDNA size distributions. Therefore, a new title is needed for the article.
2. The authors argue that their method’s main advantage over LAMA is in reducing costs of synthesizing long oligonucleotides. I would like to see this documented and quantified. Integrated DNA Technologies (IDT) offers economical G-block DNA synthesis of fragments up to 3 kbp.
3. The authors do not cite the sources of the eccDNA sequences they’ve chosen for this study.
4. There are no data regarding yield of the circular-DNA products.
5. DNA circles larger than about 500 bp would be expected to form linking-number topoisomers on ligation and hence multiple bands on an agarose gel. However even the eccLIMD1 product, which is 2.7 kbp, appears as a single band.
6. (Minor) bottom paragraph of p. 1: nM, not nm.

Author Response
1. The title of the manuscript is misleading. eccDNA sizes can range into the hundreds of kbp or even Mbp. The authors have demonstrated the utility of their method only up to 2.7 kbp, which is possibly near the average of many eccDNA size distributions. Therefore, a new title is needed for the article.
Re: Yes, some eccDNAs may have several Mbp in length. To avoid the misleading, we removed the “long” from the title of the revised manuscript.
2. The authors argue that their method’s main advantage over LAMA is in reducing costs of
synthesizing long oligonucleotides. I would like to see this documented and quantified. Integrated DNA Technologies (IDT) offers economical G-block DNA synthesis of fragments up to 3 kbp.
Re: We checked the website of IDT (https://www.geneuniversal.com/service/edit?id=14) at 2/20/2023. It shows that the price of $0.19/bp for DNA oligonucleotides in 1501bp~3000bp and waiting time of 10-14 Business days, which means researchers need spend $540 and wait two-three weeks for a 3000bp oligos. Additionally, for the oligonucleotides longer than 3000bp, IDT will charge $0.25/bp and ask 16-24 Business days to ship back. However, using QuickLAMA, researchers can spend less than $30 for 4 primers (waiting time 1-2days) and synthesis it within one day. Therefore, it is clear that our method is much cost and time efficient than the G-block DNA synthesis of IDT.
3. The authors do not cite the sources of the eccDNA sequences they’ve chosen for this study.
Re: The three eccDNAs were identified from a cancer tissue sample and PC3 cell line using round rolling circle amplification sequencing (RCA-Seq) from one of our ongoing projects and the data has not yet been published. We mentioned it in the revised paper (Table 1). Additionally, we didn’t observe significant sequencing specificity of QuickLAMA and It should work for most of genome regions.
4. There are no data regarding yield of the circular-DNA products.
Re: Thank you for the suggestions we add a new table to descript it. Please check Table 2.
5. DNA circles larger than about 500 bp would be expected to form linking-number topoisomers on ligation and hence multiple bands on an agarose gel. However even the eccLIMD1 product, which is 2.7 kbp, appears as a single band.
Re: Yes, the eccDNA products of QuickLAMA usually shown several bands or even a smear big bands in regular agarose Gel conditions, which make it difficult to evaluate the synthesis results. To overcome issue, we add 0.5% SDS to DNA loading buffer to eliminates DNA-protein interactions, prevents appearance of additional bands due to the topology diversity of DNA circles. We added the important details in the method part of the revised manuscript. Please refer to the line 203.
6. (Minor) bottom paragraph of p. 1: nM, not nm.
Re: Thank you, we fixed the error
If you have additional suggestions and comments about the manuscript, please let us know, we will try our best to improve it.
Thank you so much!
Shanru Zuo
Hunan Normal University
Changsha, Hunan, CHINA
2/26/2023

Round 2
Reviewer 1 Report
The authros foxed most of my concerns. I am fine with publishing given minor text changes. First, the abstract mentions HeLa cells, but the text says circles were derived from PC3 cells. Make sure to correctly icdentify the cell line the circles were derived from ( I asusme PC3 is correct). The authors added in the molar ration in the results section of the text but it still missing from the methods section (lines 147 and 152) and should be clarified. Editing for English spelling/grammar is still needed.
Author Response
Dear reviewer:
The authros foxed most of my concerns. I am fine with publishing given minor text changes. First, the abstract mentions HeLa cells, but the text says circles were derived from PC3 cells. Make sure to correctly icdentify the cell line the circles were derived from ( I asusme PC3 is correct). The authors added in the molar ration in the results section of the text but it still missing from the methods section (lines 147 and 152) and should be clarified. Editing for English spelling/grammar is still needed.
Many thanks for your comments and suggestions. EccMir2392 is derived from PC3 cells and remain two eccDNAs are from a cervical cancer tissue. We updated this information in the abstract in the revised manuscript. In the methodology section, we revised the related description and make it clear that it was a molar ratio. We also checked the manuscript carefully and fixed several grammar errors in the manuscript.
If you have additional suggestions and comments about the manuscript, please let us know, we will try our best to improve it.
Thank you so much!
Shanru Zuo
Hunan Normal University
Changsha, Hunan, CHINA
3/11/2023

Reviewer 3 Report
I still do not understand why only a single band is observed instead of topoisomers. Adding 0.5% SDS to the loading dye will not have any effect on topoisomer mobilities. Moreover, I believe that the authors should clearly state that these molecules are mimics of eccDNAs since the in-vitro synthesized molecules may not be biologically identical to their naturally occurring counterparts.
Author Response
Comments: I still do not understand why only a single band is observed instead of topoisomers. Adding 0.5% SDS to the loading dye will not have any effect on topoisomer mobilities. Moreover, I believe that the authors should clearly state that these molecules are mimics of eccDNAs since the in-vitro synthesized molecules may not be biologically identical to their naturally occurring counterparts.
Re: Many thanks for your comments and suggestions. After carefully checking the agarose gels, we did observe an additional band for all eccDNAs. We have attached additional agarose gel images of the experiment repeats, which clearly label these additional bands. The reason we only saw the single band in the figures 3 is that the additional bands are too weak (eccBRCA1 and eccLIMD1) or merged togather (eccMir2329)(Please see th attachement). Additionally, following your suggestions, we added a statement to highlight the possible difference between the products of QuickLAMA and their naturally occurring counterparts in the limitation section. Please refer to the line 304-307.
If you have additional suggestions and comments about the manuscript, please let us know, we will try our best to improve it.
Thank you so much!
Shanru Zuo
Hunan Normal University
Changsha, Hunan, CHINA
3/12/2023
